

# Progesterone influences cytoplasmic maturation in porcine oocytes developing *in vitro*

Bao Yuan[1,2,*], Shuang Liang[2,*], Yong-Xun Jin[1,2], Jeong-Woo Kwon[2], Jia-Bao Zhang[1] and Nam-Hyung Kim[1,2]

[1] Department of Laboratory Animal, College of Animal Sciences, Jilin university, Changchun, Jilin, P.R.China

[2] Department of Animal Sciences, Molecular Embryology Laboratory, Chungbuk National University, Cheongju, Chungbuk, Korea

[*] These authors contributed equally to this work.

## ABSTRACT

Progesterone (P4), an ovarian steroid hormone, is an important regulator of female reproduction. In this study, we explored the influence of progesterone on porcine oocyte nuclear maturation and cytoplasmic maturation and development *in vitro*. We found that the presence of P4 during oocyte maturation did not inhibit polar body extrusions but significantly increased glutathione and decreased reactive oxygen species (ROS) levels relative to that in control groups. The incidence of parthenogenetically activated oocytes that could develop to the blastocyst stage was higher ($p < 0.05$) when oocytes were exposed to P4 as compared to that in the controls. Cell numbers were increased in the P4-treated groups. Further, the P4-specific inhibitor mifepristone (RU486) prevented porcine oocyte maturation, as represented by the reduced incidence ($p < 0.05$) of oocyte first polar body extrusions. RU486 affected maturation promoting factor (MPF) activity and maternal mRNA polyadenylation status. In general, these data show that P4 influences the cytoplasmic maturation of porcine oocytes, at least partially, by decreasing their polyadenylation, thereby altering maternal gene expression.

## INTRODUCTION

In mammals, follicle-stimulating hormone (FSH) and luteinizing hormone (LH), which are secreted by the pituitary gland are primarily responsible for follicular growth and ovulation. Thus, FSH and/or LH are usually added to *in vitro* maturation (IVM) medium. In the pig, previous studies have shown that the addition of FSH to IVM medium can promote cumulus expansion and increase the ratio of oocytes undergoing germinal vesicle breakdown (GVBD) and/or achieving metaphase II (MII) (*Algriany et al., 2004*; *Su et al., 1999*). Furthermore, FSH stimulates cumulus cell expansion and increases progesterone concentration in the follicle in pigs (*Blaha et al., 2015*).

During follicular development, the concentrations of steroid hormones change, and ratios of progesterone and estradiol may affect oocyte maturation. In rhesus monkeys and

Corresponding authors
Jia-Bao Zhang, zjb515@126.com
Nam-Hyung Kim,
nhkim@chungbuk.ac.kr

humans, high ratios of P4 to estrogen seem to be related to high rates of embryogenesis and frequency of pregnancy (*Dumesic et al., 2003*; *Wagner et al., 2012*). High ratios of progesterone and estradiol in the IVM medium of oocytes promote developmental capacity in monkeys (*Zheng, 2007*; *Zheng et al., 2003*). Similarly, in cattle, progesterone in the maturation medium improves the frequency of development to the blastocyst stage (*Ryan, Spoon & Williams, 1992*). In porcine oocytes, adding P4 to the IVM medium accelerates meiosis resumption (*Eroglu, 1993*; *Sirotkin & Nitray, 1992*) and enhances IVM via follicular fluid and embryonic development. Mifepristone (RU486) is an $11\beta$-dimethyl-amino-phenyl derivative of norethindrone with a high affinity for progesterone (*Belanger, Philibert & Teutsch, 1981*). RU486 effectively prevents P4 receptors activity in porcine placentae, and can terminate pregnancy (*Hapangama & Neilson, 2009*). Previously, RU486 has been shown to suppress the cumulus expansion and meiotic maturation of porcine cumulus–oocyte complexes (COCs) in culture (*Shimada et al., 2004*; *Yamashita et al., 2010*).

Although studies indicate that P4 improves oocyte nuclear maturation, the underlying mechanism of progesterone's positive effect on the oocyte cytoplasm during *in vitro* maturation has not yet been determined. There are multiple ways to evaluate the quality of MII stage oocytes and early embryos. Intracellular levels of ROS and GSH are critical factors that influence oocyte maturation and subsequent embryo development (*Evans, Dizdaroglu & Cooke, 2004*; *Kang et al., 2016*). Maturation promoting factor (MPF) is the principal regulatory molecule driving meiotic progression during oocyte maturation (*Lin et al., 2014*). The poly(A) tail (PAT) length of the MPF gene influences further embryonic development (*Zhang, Cui & Kim, 2010*). Apoptosis-related genes and cell apoptotic rates reflect the quality of the blastocysts (*Han et al., 2016*).

In this study, progesterone was added during the *in vitro* maturation of pig oocytes. The beneficial effect of progesterone on oocyte quality was investigated by evaluating early embryonic development in porcine oocytes after progesterone supplementation during *in vitro* maturation. Furthermore, various functional features, such as ROS levels, GSH levels, maternal gene expression, polyadenylation levels, apoptosis levels in blastocysts, and p34cdc2 kinase activity in oocytes were evaluated and compared after progesterone supplementation.

## MATERIALS AND METHODS

This study was carried out in strict accordance with the Guide for the Care and Use of Laboratory Animals of Jilin University. Animal procedures were conducted following the protocol (20151207) approved by the Animal Care & Welfare Committee of Jilin University. All chemicals used were purchased from the Sigma-Aldrich Chemical Company (St. Louis, MO, USA) unless otherwise stated.

### Collection and IVM of porcine oocytes

Porcine COCs were recovered from follicles 3–6 mm in diameter in porcine ovaries and washed three times with TL-HEPES (with 0.05 g/L gentamycin and 1 g/L polyvinyl alcohol (PVA) added). The collected COCs were matured in IVM medium for 44 h at 38.5 °C in 5% $CO_2$ and humidified air. Normal IVM medium is comprised of tissue culture medium 199 (Gibco) supplemented with 0.1 g/L sodium pyruvate, 0.6 mM L-cysteine, 10 ng/mL

**Table 1  Primers used for PAT and real-time PCR.**

| Gene | GenBank number | Primer sequences (5′–3′) | Product size(bp) |
| --- | --- | --- | --- |
| PAT-Cdc2 | NM_001159304 | CTGTTAACTCTGCTTTTGTCTTGTGT | – |
| Oligo(dT)-Anchor | – | GCGAGCTCCGCGGCCGCGT$_{12}$ | – |
| GAPDH | NM_001206359 | F: GTCGGTTGTGGATCTGACCT | 207 |
| | | R: TTGACGAAGTGGTCGTTGAG | |
| Cdc2 | NM_001159304.2 | F: TAATAAGCTGGGATCTACCACATC | 185 |
| | | R: CGAATGGCAGTACTAGGAACAC | |
| cyclin B1 | NM_001170768.1 | F: AGCTAGTGGTGGCTTCAAGG | 101 |
| | | R: GCGCCATGACTTCCTCTGTA | |
| Bax | XM_005664710.2 | F: GGTCGCGCTTTTCTACTTTG | 111 |
| | | R: CGATCTCGAAGGAAGTCCAG | |
| Bcl2 | NM_214285 | F: AGGGCATTCAGTGACCTGAC | 193 |
| | | R: CGATCCGACTCACCAATACC | |
| Casp3 | NM_214131.1 | F: ACTGTGGGATTGAGACGG | 110 |
| | | R: GGAATAGTAACCAGGTGCTG | |

epidermal growth factor, 10% porcine follicular fluid (PFF) (v/v), 10 IU/mL LH, and 10 IU/mL FSH. We also tried using normal IVM medium that was not supplemented with PFF, LH, or FSH.

Based on a previous study (*Salehnia & Zavareh, 2013*; *Shimada & Terada, 2002*) different concentrations of progesterone (0 µM, 10 µM, or 100 µM) and RU486 (0 µM, 10 µM, or 25 µM) were added to the culture media. After IVM, the COCs were washed in TL-HEPES (with hyaluronidase (1 mg/mL) and PVA (0.1%, v/v) added) to remove cumulus cells. The oocytes were added to normal TL-HEPES and the oocytes in which the first polar bodies had discharged were selected for further studies.

### Measurement of MII oocyte ROS and GSH levels

To detect the ROS and GSH levels, MII stage oocytes were sampled in medium with added P4 (100 µM) or RU-486 (25 µM) for determination of their intracellular ROS and GSH levels. For detection of the ROS levels, the oocytes were incubated with 10 µM $H_2DCFDA$ for 15 min (green fluorescence, UV filters, 460 nm). For detection of the GSH levels, the oocytes were incubated with 10 µM CMF2HC (Invitrogen) for 15 min (blue fluorescence, UV filters, 370 nm). The same procedures were followed for all groups of oocytes, including incubation, rinse, mounting and imaging. Image J software was used to analyze the fluorescence intensities of the oocytes. Three independent experiments were performed.

### Analysis of poly(A) tail lengths by polymerase chain reaction (PCR)

To detect the maternal transcripts poly(A) tail length, PAT assay was performed as described previously (*Zhang, Cui & Kim, 2010*). Briefly, total mRNAs from MII pig oocytes were reverse transcribed with anchored oligo(dT) primer (Table 1) (*Salles & Strickland, 1999*). PCR was performed using anchored oligo(dT) primer and PAT-Cdc2 primer to test the maternal transcripts (Table 1). The adjustment PCR program was run for 5 min at 95 °C,

followed by 35 cycles of 20 s at 94 °C, 45 s at 60 °C, 45 s at 72 °C, and finally, for an extension of 3 min at 72 °C. A 3.0% agarose gel electrophoresis was performed to analyze the PCR products.

## MII oocyte MPF activity assay

The Cdc2/Cdk1 Kinase Assay Kit (MBL, Nagoya, Japan) was used to quantify p34cdc2 kinase activity (*Lin et al., 2014*; *Zhang, Cui & Kim, 2010*). Briefly, 30 oocytes were washed three times in sample buffer. Oocyte extract (5 µL) was mixed with kinase assay buffer (45 µL). The mixture was placed in an incubator at 30 °C for 30 min. The reaction was terminated by 200 µL ethylene glycol tetraacetic acid (50 mM). The OD value was read at 492 nm. Three independent experiments were performed.

## Parthenogenetic activation and *in vitro* culture of pig oocytes

After 42 h of maturation, MII stage oocytes were selected. Denuded oocytes with homogeneous cytoplasm were selected and then gradually equilibrated in activation solution by a 1.0 kV/cm electric pulse for 60 µs. Activated oocytes were hatched in PZM-5 medium with 2 mM cytochalasin B for 3 h. Next, approximately 40–50 post-activation oocytes were cultured in PZM-5 for 7 days, and embryos were cultured at 38.5 °C in 5% $CO^2$ (*Kwon, Namgoong & Kim, 2015*).

## Real-time reverse transcription PCR

Total mRNA extraction and cDNA synthesis were performed as previously described (*Lee et al., 2014*). Briefly, 50 MII oocytes or 20 blastocysts were used to extract mRNA with a Dynabeads mRNA Direct Kit, following reverse transcription of the mRNA by oligo(dT) 12–18 primer with a SuperScript Reverse Transcriptase Kit. The primers used for the real-time PCR (RT-PCR) are listed in Table 1. The reaction was performed in a Bio-Rad CFX PCR machine. The $2^{-\Delta\Delta Ct}$ method was followed to analyze gene expression (*Livak & Schmittgen, 2001*). The control gene was glyceraldehyde 3-phosphate dehydrogenase (*GAPDH*). Three independent experiments were performed in triplicate.

## Confocal microscopy and counting the number of nuclei per blastocyst

Counting methods were described previously (*Liang et al., 2015*). Briefly, the blastocysts were fixed in 3.7% paraformaldehyde prepared in PBS-PVA for 30 min, then washed in PBS washing solution and 0.3% Triton X-100 for 1 h. After being washed twice in PBS, the blastocysts were put in fluorescein-conjugated dUTP and terminal deoxynucleotidyl transferase enzyme in the dark for 1 h. Next, they were incubated with 10 µg/mL Hoechst 33342 and 50 mg/mL RNase A for 1 h. Finally, the blastocysts were examined under a laser scanning confocal microscope.

## Statistical analyses

All the experiment results were analyzed by one-way ANOVA and a Chi-square test to determine the *p*-value, using IBM SPSS Statistics 19 software. A *p*-value < 0.05 was considered statistically significant.

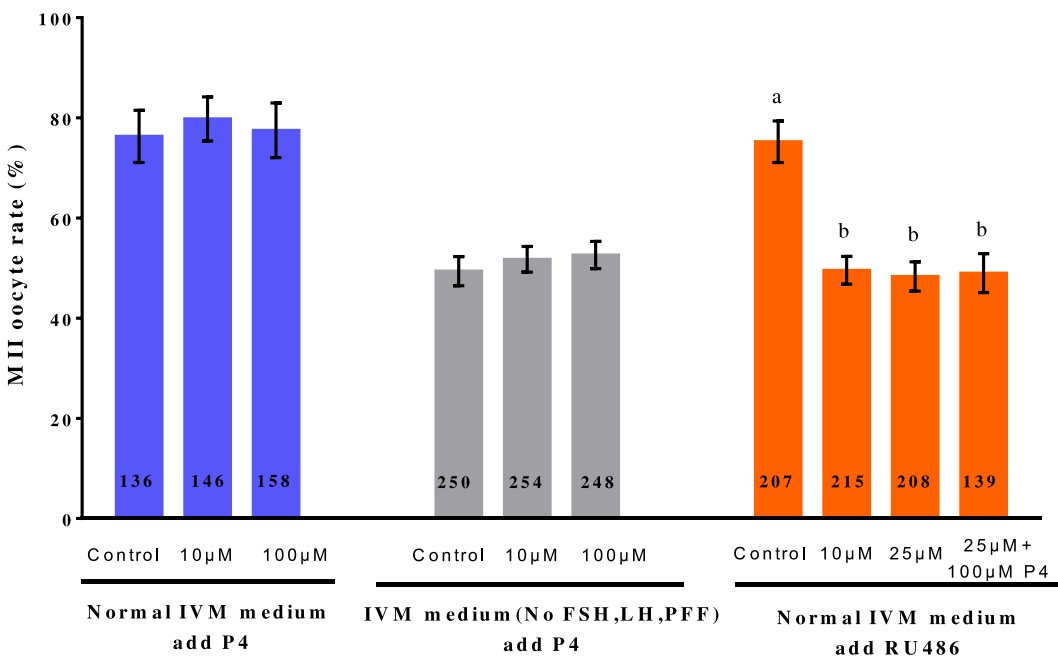

**Figure 1** **Effect of adding P4 or RU486 on IVM of porcine oocytes.** The number of oocytes observed in each experimental group is displayed in the bar. Bars with different superscripted letters (a or b) in each column indicate statistically significant differences ($p < 0.05$). Values shown are the mean ($\pm$ standard deviation) of three independent experiments.

## RESULTS

### Effects of P4 and RU486 IVM on porcine oocytes

To evaluate the effects of P4 and RU486 on porcine oocyte maturation, porcine oocytes were treated with various concentrations of P4 (0, 10, or 100 $\mu$M) and RU486 (0, 10, or 25 $\mu$M), and their polar body extrusions were examined. The oocytes were randomly divided into three groups: normal IVM medium with added P4; IVM medium (without FSH, LH, and PFF) with added P4; and normal IVM medium with added RU486 (Fig. 1). In both the normal IVM medium and IVM medium (without FSH, LH, and PFF) groups, adding different concentrations of P4 had no effect on MII oocyte rate ($p > 0.05$). However, in the normal IVM medium with added RU486 group, adding 10 $\mu$M or 25 $\mu$M RU486 significantly reduced the MII oocyte rate ($p < 0.01$).

### Effects of P4 and RU486 on intracellular levels of ROS and GSH

To determine the mechanism by which P4 and RU486 influence porcine oocyte maturation, the levels of ROS and GSH were examined. Significant decreases in the levels of ROS and GSH were observed in porcine oocytes after IVM. The levels of ROS were significantly lower in the 100 $\mu$M P4-treated oocytes (4.95 $\pm$ 1.57 pixels/oocyte; Fig. 2B), but higher in the 25 $\mu$M RU486-treated oocytes (15.48 $\pm$ 3.67 pixels/oocyte; Fig. 2C) than in the control group (8.65 $\pm$ 1.99 pixels/oocyte; Fig. 2A). The levels of GSH were higher significantly in the 100 $\mu$M P4-treated oocytes (69.29 $\pm$ 4.81 pixels/oocyte; Fig. 2E), but lower in the 25 $\mu$M RU486-treated oocytes (47.40 $\pm$ 8.28 pixels/oocyte; Fig. 2F) than in the control group (63.78 $\pm$ 7.05 pixels/oocyte; Fig. 2D).

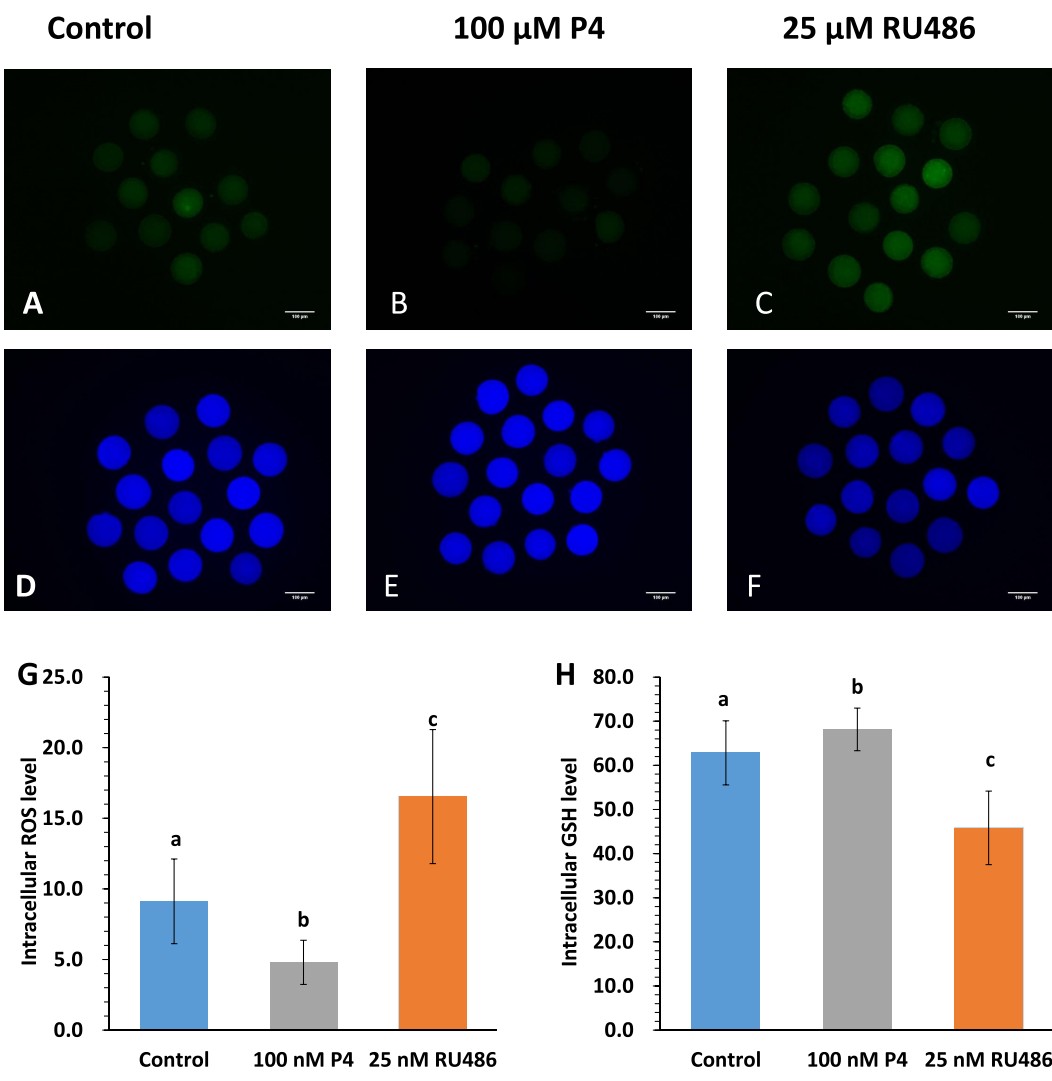

**Figure 2  ROS and GSH images of MII oocytes.** Representative images showing ROS and GSH expression in MII oocytes. Oocytes were dyed with $H_2$DCFDA (A–C) and Cell Tracker Blue (D–F) to detect ROS levels and GSH levels. MII oocytes from the control IVM medium and 100 $\mu$M P4- or 25 $\mu$M RU486-supplemented IVM system. Effects of P4 or RU486 supplementation during IVM on intracellular ROS and GSH levels in mature oocytes (G, H). Bars with different superscripted letters (a, b, and c) in each column indicate statistically significant differences (GSH or ROS; $p < 0.05$). The experiment was replicated three times.

## Effects of P4 and RU486 on maternal gene expression, polyadenylation levels, and p34cdc2 kinase activity

Maternal gene expression is an important biological process in oocyte maturation and early embryo development. We examined the expression of maternal genes *cdc2* and *cyclinb1* (regulatory subunits of MPF). After treatment of the oocytes with P4, their *cdc2* and *cyclinb1* mRNA levels increased, but decreased after 44 h of treatment with RU486 (Fig. 3A). We also analyzed the *cdc2* gene poly(A) tail length at the MII stage (Fig. 3B). In this treatment, 25 $\mu$M RU486 affected the maternal mRNA polyadenylation status by shortening signal

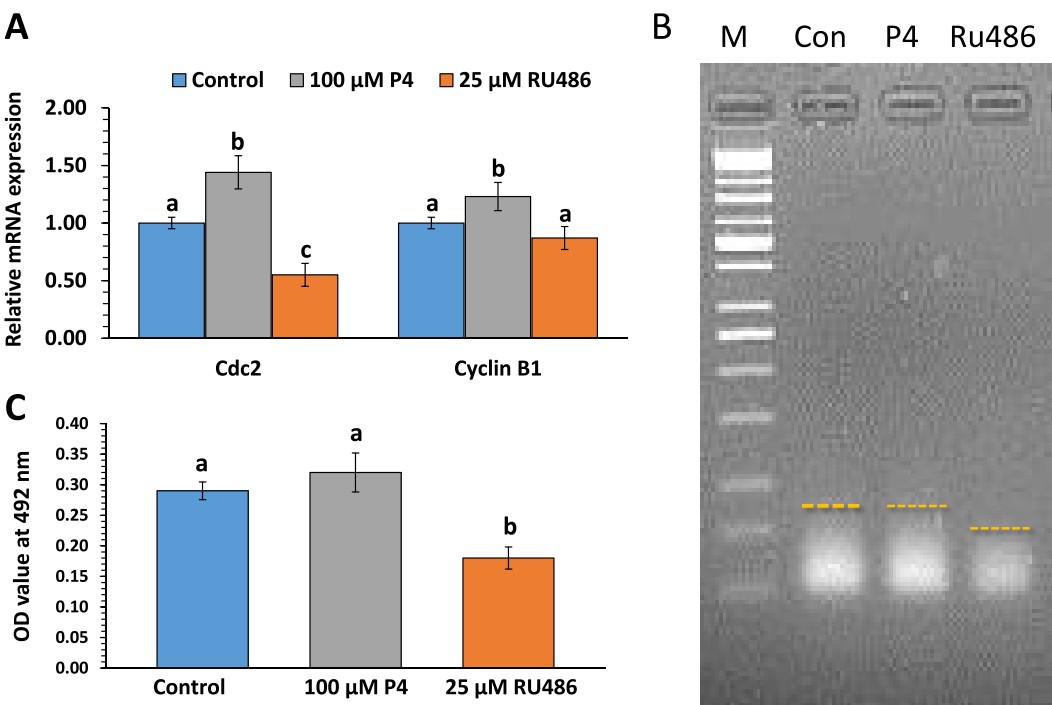

**Figure 3** **The effects of P4 or RU486 on maternal gene expression, polyadenylation levels and p34cdc2 kinase activity.** Maternal mRNA expression at the MII stage (A). Analysis of poly(A) tail length of *Cdc2* transcripts at the MII stage (B). Treatment with 100 μM P4 or 25 μM RU486 are indicated at the top of the figure. Oocytes at the MII stage (44 h). Differences in poly(A) tail lengths of maternal mRNA represented by PCR smears are indicated by dotted lines. MPF activity in MII oocytes (C). MPF was isolated from oocytes treated with 100 μM P4 or 25 μM RU486. Data are expressed as the percentage ± SEM of three independent replicates of three experiments. Different superscripted letters show significance ($p < 0.05$).

smears. In 100 μM P4-treated oocytes, *cdc2* underwent polyadenylation at MII; there were no significant differences from that in the control group. When COCs were IVM-treated with RU486, the activation of *p34cdc2a* decreased at the MII stage as compared with that in the control and P4treated groups (Fig. 3C).

## Effects of P4 and RU486 during IVM on embryo development

To determine whether P4 or RU486 treatment during IVM influences subsequent embryonic development, porcine oocytes were activated and their *in vitro* development examined (Fig. 4). In normal medium, the addition of different concentrations of P4 had no effect on the oocyte blastocyst rate ($p > 0.05$). In medium without FSH, LH, and PFF, the blastocyst rate was significantly higher in the 100 μM P4-treated group than in the other groups (50.56% vs. 42.30%, 38.36%; $p < 0.05$). However, adding 10 μM or 100 μM RU486 significantly reduced the blastocyst rate (34.44%, 32.26% vs. 47.16%).

## TUNEL assay at the blastocyst stage

DNA fragments generated by apoptotic nicking of genomic DNA were measured in individual embryos by TUNEL assay. The apoptotic rate at the blastocyst stage was significantly higher in the RU486-treated group than in the P4-treated group (Figs. 5A

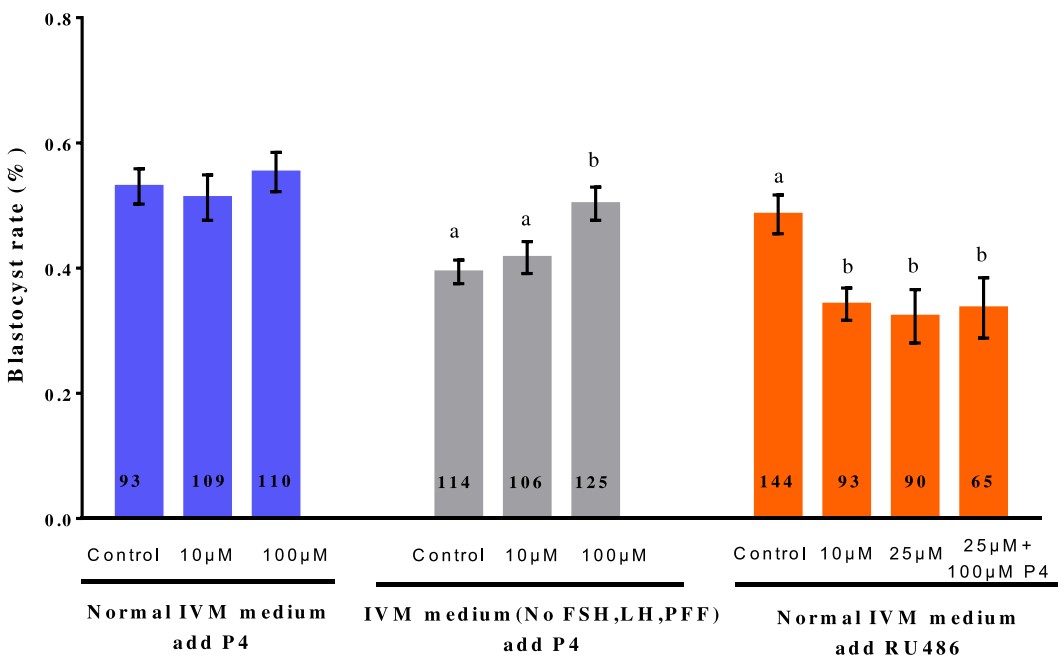

**Figure 4 Effect of treatment with P4 or RU486 during IVM of porcine oocyte development to the blastocyst stage.** The number of oocytes observed in each experimental group is displayed in the bar. Different letters indicate $p < 0.05$.

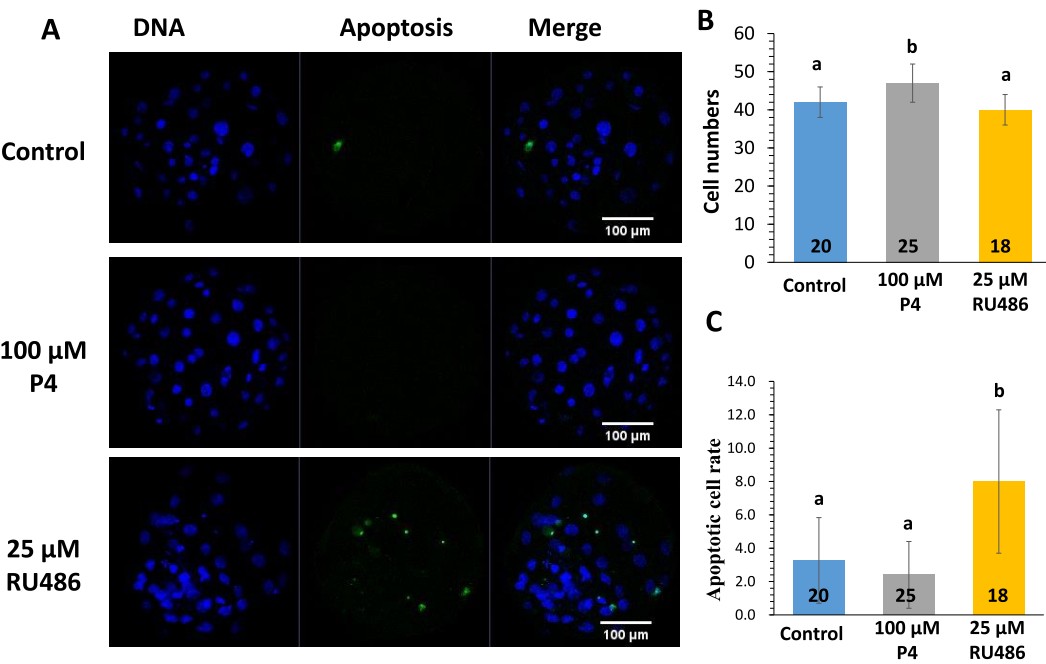

**Figure 5 TUNEL assay at blastocyst stage.** Representative confocal images of apoptotic cells and nuclear DNA ($\times 400$) (A). Total numbers of cells per blastocyst in different groups (B). Apoptotic rate per blastocyst in different groups (C). The number of oocytes observed in each experimental group is displayed in the bar. Different letters indicate $p < 0.05$.

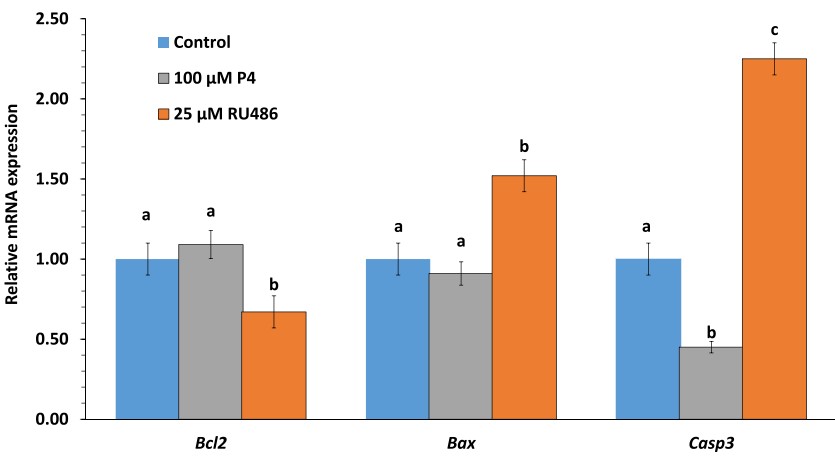

**Figure 6   Effects of P4 or RU486 in IVM on blastocysts apoptosis-related gene expression.** *Bcl-2*, *Bax*, and *Casp3* mRNA expression levels in 7-day porcine blastocysts with P4 or RU486 treatment during IVM. In the bars, different letters indicate $p < 0.05$.

and 5C). The number of blastocysts was higher in the P4-treated group than in the control group ($42.1 \pm 3.9$ vs. $47.5 \pm 5.3$); however, no significant difference was found between the RU486-treated and control groups (Fig. 5B). To further explore the manner in which P4 and RU486 influence the incidence of apoptotic cells in parthenogenetic blastocysts, the expression of the apoptosis-related genes *Bcl2*, *Bax*, and *Casp3* was evaluated in parthenogenetic blastocysts. Compared with that in the non-treated group, the expression of *Casp3* and *Bax* mRNA was significantly higher and the expression of *Bcl-2* mRNA was significantly lower in the RU486-treated group (Fig. 6).

## DISCUSSION

Different maturation media have been reported to have distinct effects on oocyte IVM (*Wang et al., 1997*), and oocytes are more susceptible to compromised developmental potency under suboptimal conditions (*Zhou et al., 2012*). Previous studies have shown that the level of progesterone produced by cumulus cells is increased by the stimulation of LH and FSH and influences porcine oocyte maturation (*Shimada & Terada, 2002*). However, the mechanism by which P4 influences porcine oocyte maturation is not clear. In this study, we examined the effects of P4 and RU486 on IVM of porcine COCs. Adding P4 to normal culture medium did not improve COC maturation and developmental ability. Adding P4 to the culture medium without PFF, LH, and FSH did not improve the maturation of porcine COCs, but did increase their developmental ability when 100 µM P4 was added. Adding 10 µM or 25 µM RU486 significantly reduced the maturation rate of porcine oocytes and their developmental ability.

Studies indicate that progesterone is produced by the cumulus cells during IVM (*Shimada & Terada, 2002*), and co-culture with cumulus cells significantly increases maturation rate and blastocyst formation of denuded oocytes during IVM in sheep (*Kyasari et al., 2012*), mouse (*Jiao et al., 2013*), porcine (*Yoon et al., 2015*), goat (*Wang et al., 2011*), human (*Combelles et al., 2005*), and rat (*Jiao et al., 2016*). However, it is not clear whether

progesterone produced by the cumulus cell monolayer is also positively involved in these co-culture systems. Our results agreed with those of previous reports on the function of P4 in bovine oocyte maturation (*Aparicio et al., 2011*). The results appear to show a positive role for P4 in MII-stage oocyte quality. P4 also improves *in vitro* cytoplasmic maturation in monkey and canine oocytes (*Vannucchi et al., 2006*; *Zheng et al., 2003*). In contrast to the results of the present study, adding P4 to the medium did not enhance the maturation of mouse germinal vesicle (GV) oocytes or their developmental ability (*Zavareh, Saberivand & Salehnia, 2009*). P4 supplementation of IVM culture systems did not affect the rate of IVM of bovine oocytes, and addition of P4 to the IVF medium did not improve the rate of cleavage stage embryos (*Carter et al., 2010*). Our results were similar to those found in porcine, bovine, and primate oocytes: P4 didn't increase the percentages of oocyte maturation (*Karlach, 1986*; *Nagyova et al., 2014*; *Ryan, Waddington & Campbell, 1999*; *Zheng et al., 2003*). However, the progesterone antagonist (RU486) could not reverse the repression effect of progesterone on mouse oocyte maturation (*Zavareh, Saberivand & Salehnia, 2009*).

ROS are known to be mediators of caspase-dependent cell death (*Jang et al., 2013*; *Le Bras et al., 2005*). Previous studies have demonstrated that progesterone possesses antioxidant properties, forming scavenging rings of ROS in cancer cells and increasing superoxide dismutase (SOD) activity in human endometrial stromal cells (*Matsuoka et al., 2010*; *Nguyen & Syed, 2011*). Progesterone (0.5 μM) protects mouse pancreatic islets against $H_2O_2$-induced oxidative stress and leads to decreased ROS production (*Ahangarpour et al., 2014*). In addition, other studies show that P4 has the ability to increase ROS levels in MCF-7 cells (*Azeez et al., 2015*). However, ROS signaling does not show any apoptotic effect of RU486 treatment in U937 cells (*Jang et al., 2013*). It has been shown that mitochondrial dysfunction is directly responsible for impaired developmental potential of oocytes (*Dai et al., 2015*). RU486, a P4 receptor antagonist, almost completely blocked the effect of progesterone on ROS protection, indicating that ROS overproduction is mediated via GC receptors.

Adding an antioxidant to the medium influences maturation rate and developmental ability. Previous studies on cattle, goats, and pigs have shown that antioxidants improve embryo developmental ability and increase embryonic GSH levels (*Droge, 2002*; *Mukherjee et al., 2014*). In the present study, GSH activity in MII oocytes was decreased significantly when RU486 was added during IVM.

MPF signals are critical for oocyte maturation. Moreover, *cdc2* and *cyclinB1* are important genes of maturation promoting factor (*Zhao et al., 2014*). In previous studies, we found that in good-quality MII-stage oocytes, *cyclinb1* and *cdc2* gene expression was increased (*Lin et al., 2014*; *Zhang, Cui & Kim, 2010*). In this study, we used P4-treated oocytes that had high expression of the *cdc2* gene. Examination of poly(A) clearly showed degradation of Cdc2 isoforms in MII oocytes (*Zhang, Cui & Kim, 2010*). In the present study, progesterone had no significant effect on the length of poly(A), and after RU486 treatment, poly(A) length decreased. Previous studies reported changes in MPF expression in porcine oocytes as they reactivated to enter MII (*Lin et al., 2014*). In the present study, the decline in MPF activity occurred following RU486 treatment.

In previous studies, high-quality oocytes showed lower expression levels of *Bax* mRNA, but higher levels of *Bcl-2* mRNA (*Yang & Rajamahendran, 2002*). In the present study, we found that RU486 decreased oocyte quality and increased apoptosis in blastocysts; this is similar to the results of other studies where RU486 was used to promote susceptibility to apoptosis (*Quirk, Cowan & Harman, 2004*). We hypothesize that the effect of P4 is to inhibit blastocyst apoptosis, but treatment with P4 only increases the number of blastocysts.

In conclusion, we focused on the effects of P4 on porcine oocyte maturation. Our results support the hypothesis that addition of P4 improves the viability of porcine oocytes and their *in vitro* developmental ability at least partially by decreasing their polyadenylation, thereby altering the expression of other maternal genes.

### Funding

This study was supported by the BioGreen 21 Program (No. PJ011126) and the National Natural Science Foundation of China (31572400). The funders had no role in study design, data collection and analysis, decision to publish, or preparation of the manuscript.

### Grant Disclosures

The following grant information was disclosed by the authors:
BioGreen 21 Program: PJ011126.
National Natural Science Foundation of China: 31572400.

### Competing Interests

The authors declare there are no competing interests.

### Author Contributions

- Bao Yuan and Shuang Liang conceived and designed the experiments, performed the experiments, analyzed the data, contributed reagents/materials/analysis tools, wrote the paper, prepared figures and/or tables.
- Yong-Xun Jin and Jeong-Woo Kwon performed the experiments, contributed reagents/materials/analysis tools.
- Jia-Bao Zhang and Nam-Hyung Kim conceived and designed the experiments, wrote the paper, reviewed drafts of the paper.

### Animal Ethics

The following information was supplied relating to ethical approvals (i.e., approving body and any reference numbers):

This study was carried out in strict accordance with the Guide for the Care and Use of Laboratory Animals of Jilin University. Animal procedures were conducted following the protocol (20151207) approved by the Animal Care & Welfare Committee of Jilin University.

## Data Availability

The raw data has been supplied as Supplementary Files.

## Supplemental Information

Supplemental information for this article can be found online at http://dx.doi.org/10.7717/peerj.2454#supplemental-information.

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
