# Peer review of "Progesterone influences cytoplasmic maturation in porcine oocytes developing in vitro"

_PeerJ, doi:10.7717/peerj.2454_

## Round 0.1 · original submission · Major Revisions

As you can see, both reviewers showed interest in your results, however, the reviewers, especially Reviewer 2, stated that your manuscript is poorly written and a complete re-write of the manuscript is needed to improve readability and resolve the English language problems. You also need to respond to the concerns the reviewers raised.

Reviewer 1 ·

Basic reporting

No Comments.

Experimental design

No Comments.

Validity of the findings

No Comments.

Additional comments

In the manuscript entitled “Progesterone influences cytoplasmic maturation in porcine oocytes developing in vitro”, the authors investigated the effects of progesterone on porcine oocyte maturation, focusing on the events of oocyte maturation, preimplantation embryo development, redox, apoptosis and expression of maternal genes. Overall, this manuscript is generally well written but with some minor grammatical errors. The results are clearly demonstrated and technically sound as well. It is also very interesting and significant for improving mammalian oocyte maturation system and embryo production. However, small issues still exist based on the current manuscript and comments are outlined below.

1. Fig. 1, how were the added concentrations of P4 and RU486 decided? It seems more appropriate to point out the context.

2. Please add scales to Fig. 2 A-F, Fig. 6A.

3. Regarding assessment of ROS and GSH levels, since only single channel was evaluated as the final result, authors need specify whether different groups were treated exactly the same, including incubation, rinse, mounting and imaging.

4. Fig. 3 legend needs revision.

5. Please pay attention to nomenclature. mRNA should be italic (including Figure legends and Figure x-axis titles).

6. Fig. 6A, no quantification was demonstrated, so Line 192 needs revision.

7. Line 59 needs reference(s).
Line 125, reference needs correction.

8. Please update RU486 full name (Line 58). This sentence also needs correction.

9. Different maturation media were reported to have distinct effects on oocyte IVM (PMID: 9370973), and oocytes are more susceptible to the compromised developmental potency under suboptimal conditions (PMID: 22133696). In this study, authors detected the similar and intriguing results, however, related explanation/discussion is insufficient. Especially, previous study also found the level of progesterone produced by the cumulus cells is increased by stimulation of LH and FSH (PMID: 12087075). It seems authors need to add these discussions and references into the manuscript (first paragraph of Discussion part).

10. Authors mentioned the controversy regarding the role of P4 in oocyte IVM, but they were mainly focused on COC system. Actually, it is well established that progesterone is produced by the cumulus cells during IVM (PMID: 12087075), and recent results did confirm that co-culture with cumulus cells significantly increased maturation rate and blastocyst formation of denuded oocytes during IVM in sheep (PMID: 21855989), mouse (PMID: 23469259), porcine (PMID: 25442018), goat (PMID: 21453051), human (PMID: 15695316), and rat (PMID: 26679437). Whether progesterone produced by the cumulus cell monolayer is also positively involved in these co-culture systems? Authors are suggested to add this discussion and cite these references to attract more readers and broaden the audience.

11. The first sentence (Line 42-44) of Introduction needs revision.
Line 45, "add" and "in order to" need replacements.
Please double check the meaning of Line 57.
Line 66 "improved" and Line 68 "supplemented" need correction.
Line 86-89 needs revision.
Paragraph of “Parthenogenetic activation and in vitro culture of pig oocytes” needs revision.
Line 128 needs revision.
Line 138 and 140 need revision.
Line 147 needs correction.
Line 168-169 needs correction.
Typo: “This rswults”.

Reviewer 2 ·

Basic reporting

No Comments

Experimental design

No Comments

Validity of the findings

No Comments

Additional comments

In this manuscript, the authors showed that the addition of progesterone in maturation medium improved cytoplasmic maturation in pig oocytes. Moreover, this beneficial effect of progesterone might be through the regulation of ROS generation, polyadenylation of mRNAs and subsequent gene expression and apoptosis. Overall this work is potentially interesting, but MS is poorly written and organized. Sometimes, it is difficult to read and follow the rationale of the work. Worse, there are numerous incorrect sentences and references. The authors need extensive editing of MS before submission.

1. Enrich the introduction by introducing the effects of ROS, GSH, maternal gene expression, poly A tailing and cellular apoptosis on oocytes maturation and subsequent embryo development.

2. Please define IVM medium.

3. RU-486 targets glucocorticoid receptor as well as P4 receptor. Because glucocorticoid has been shown to affect the cytoplasmic maturation of porcine oocytes, it could not exclude the possibility that RU-486 may influence the action of glucocorticoid which may present in porcine follicular fluid (PFF) on oocyte maturation. Can authors comments on this?

4. Please explain the rationale of each experiment at the beginning of sub-section. For example, in fig.2, why do they determine ROS and GSH level? Also, why do they examine poly A length of cdk1?

5. Fig. 4. P4 had no effect on IVM when cultured without FSH, LH and PFF. However, blastocyst formation was increased when activated. Explain why?

6. Throughout the experiments, P4+RU-486 group should be included.

---

## Round 0.2 · accepted · Accept

Both reviewers stated that your revision addressed their concerns and therefore I recommended the publication of your manuscript in PeerJ.

Reviewer 1 ·

Basic reporting

No Comments.

Experimental design

No Comments.

Validity of the findings

No Comments.

Additional comments

All concerns and issues have been well addressed.

Reviewer 2 ·

Basic reporting

No comments

Experimental design

No comments

Validity of the findings

No comments

Additional comments

The authors have adequately addressed all my concerns, and I support the publication of the current version of MS.